# Effectiveness of an Online Dementia Prevention Program on Cognitive Function and Depression in Community-Dwelling Older Adults during the COVID-19 Pandemic in Korea

**DOI:** 10.3390/healthcare11101376

**Published:** 2023-05-10

**Authors:** Kyoung-Chul Min, Eun-Hee Kim, Hee-Soon Woo, Chiang-Soon Song

**Affiliations:** 1Department of Occupational Therapy, Seoul Metropolitan Childern’s Hospital, Hunneungro 260, Seochogu, Seoul 60801, Republic of Korea; 2Department of Occupational Therapy, Uiwang city Public Health Center, Obongro 34, Uiwang, 16075 Gyeonggi-do, Republic of Korea; 3Department of Occupational Therapy, School of Medicine, Wonkwang University, 460 Iksandae-ro, Iksan, Jeonbuk 54538, Republic of Korea; 4Department of Occupational Therapy, College of Natural Science and Public Health and Safety, Chosun University, Chosundae-5gil, Dong-gu, Gwangju 61452, Republic of Korea

**Keywords:** community-based occupational therapy, daily task participation, dementia prevention program, online dementia prevention program, online cognitive therapy

## Abstract

(1) Background: Continuous participation in a comprehensive dementia prevention program is important for community-dwelling older adults during the coronavirus disease (COVID-19) pandemic, as limitations on their communities and social participation have increased and participation in daily tasks has decreased. These factors can negatively affect their cognitive function and symptoms of depression. This study aimed to introduce an evidence-based online dementia prevention program in the South Korean context and to identify its effect on cognitive function and symptoms of depression in community-dwelling older adults during the COVID-19 pandemic. (2) Methods: One hundred and one community-dwelling older adults without dementia participated in twelve sessions of an online dementia prevention program designed by occupational therapists. Cognitive function and symptoms of depression were assessed before and after the program. Cognitive function was tested using the Cognitive Impairment Screening Test and symptoms of depression were assessed using the Korean version of the Short Geriatric Depression Scale. The participants’ opinions were gathered using open-ended questions. (3) Results: After the program, according to the raw score, orientation was maintained and attention, visuospatial function, executive function, memory, and language function increased. The memory and total cognitive score was improved significantly. Symptoms of depression significantly decreased. The program’s benefits according to the participants were participation in new activities, boredom reduction, online communication, and reminiscence. (4) Conclusions: An online dementia prevention program is effective in maintaining and increasing cognitive function and preventing depression in community-dwelling older adults. An online dementia prevention program is a useful method in providing opportunities to participate in cognitive training and continuous daily activities during the COVID-19 pandemic.

## 1. Introduction

Face-to-face rehabilitation was commonly used before the coronavirus disease (COVID-19) pandemic. However, during the COVID-19 pandemic, community-dwelling older adults have limited access to public health services and limited participation in classic face-to-face rehabilitation and community treatment services [1,2]. These situations negatively affected the motor, cognitive, and psychosocial functions of older adults [3,4]. In particular, older adults need continuous rehabilitation, participation in daily tasks, and communication with others to maintain their functions and mental health [2,5]. Moreover, older adults are particularly vulnerable to the COVID-19 pandemic because they have lost many opportunities available to them [5,6].

Online rehabilitation is a useful treatment strategy during the COVID-19 pandemic. Continuous participation in rehabilitation, cognitive therapy, and community services are essential for the older adults and those with mild cognitive impairment (MCI), dementia, and stroke [6]. Participation in comprehensive daily activities such as keeping a diary has positive effects on cognitive function, independent living, participation, mental health, and quality of life [3,5,7]. Recently, dementia prevention and management for community-dwelling older adults have become a major part of the local health system [7]. The effectiveness of many types of online cognitive and dementia prevention programs has been identified in previous studies. 

Online dementia prevention programs are effective in maintaining cognitive function, and in preventing and delaying cognitive impairment in older adults with MCI and dementia [7,8,9,10]. Online dementia prevention programs included participation in daily activities, keeping a diary, and using online cognitive programs. Researchers have concluded that daily task participation is effective in improving cognitive function in older adults. This improvement is also achieved in online treatment settings. 

Online rehabilitation therapy is a useful method for managing depression in community-dwelling older adults. Communication with others can decrease depression levels [11,12]. Therefore, a lack of participation in community activities is significantly correlated with depression [11]. Depression affects social roles, health, and social well-being [12]. Additionally, daily activities such as keeping a diary may decrease the depressive symptoms of chronic rehabilitation patients [9].

Evidence-based online dementia prevention programs are important. The American Psychiatric Association (APA) suggested a comprehensive dementia prevention program that includes behavioral, emotional, cognitive, and stimulus-oriented approaches [13]. Therefore, to develop a proper dementia prevention program, specialists could consider various areas including four factors as described. Thus, experts such as occupational therapists (OTs) should play a major role in developing and conducting comprehensive dementia programs. Comprehensive treatment includes cognitive training and periodic cognitive activities such as diary writing, daily occupational tasks, and activities [14,15,16,17]. Furthermore, according to the occupational therapy theory, occupational participation allows individuals to engage in the environment, perform occupational activities, and maintain their occupation [18].

Previous studies have demonstrated the effectiveness of online cognitive rehabilitation in older adults and those with MCI and dementia. In South Korea, face-to-face dementia prevention programs are usually conducted at public health centers. In particular, OTs play a major role in designing and performing dementia prevention programs for community-dwelling older adults in the Korean context. Dementia prevention programs are a major element in Korean public health centers. Therefore, sharing and introducing ideas of evidence-based dementia prevention programs is important. However, studies about identifying the effectiveness of online dementia prevention programs through participation in daily activities and cognitive activities for healthy older adults in a community environment are insufficient.

The aims of this study were to introduce an online dementia prevention program based on the APA guidelines and to identify the effect of online dementia prevention programs on cognitive function and depression in community-dwelling older adults in the Korean context. 

## 2. Materials and Methods

### 2.1. Subjects

One hundred and one community-dwelling older adults participated in this study at a public health center in Gyeonggi-do, Korea. Before the program, they received a sufficient explanation about their participation in the dementia prevention program and research. All participants provided written informed consent. The inclusion criteria were age >60 years, normal cognitive function, mild cognitive impairment (MCI), ability to use smartphones, ability to read and write, Internet installation at home, and no hearing or visual limitations. The exclusion criteria were dementia, difficulties in using smartphones, inability to read and write, hearing and visual limitations, and no Internet connection at home. There were no dropouts in the middle of the program.

### 2.2. Procedures

The online dementia prevention program was conducted once a week, 1 hour a day, for 12 weeks (12 sessions) from April to June 2021. All community activities and gatherings were banned by the Korean government during the study period to protect against the spread of COVID-19. This program was designed by two authors (first and second) based on the APA guidelines. The second author recorded all program videos, which were 30–40 min long, and uploaded them on the NAVER BAND application (app) once a week. NAVER BAND app is a commercial communication app that allows subscribers to communicate and upload files. Participants watched the program videos and performed daily tasks by keeping a diary and a brain health book. Program videos were uploaded to the NAVER BAND app during the program period. They watched the videos whenever they wanted during the program. Subsequently, they uploaded their weekly tasks to the NAVER BAND app. The authors communicated with participants through text messages. Before and after the program, cognitive functions and symptoms of depression were assessed. 

Participants visited a public health center before the start of the program. All participants were retired. Cognitive function and symptoms of depression were assessed directly by the second author. Education and training on how to use the NAVER BAND app were provided to participants and their caregivers. After the program, participants visited a public health center to be assessed for cognitive function and symptoms of depression directly. Every participant uploaded their tasks on time. 

### 2.3. Outcome Meausres

#### 2.3.1. Cognitive Impairment Screening Test (CIST) 

The CIST is a cognitive function screening test developed by the Ministry of Health and Welfare and the Central Dementia Center in South Korea [19]. It consists of 13 items including 6 domains of orientation, attention, visuospatial function, executive function, memory, and language function. Questions included identifying the date, day, month, and year, repeating sentences, saying numbers backward, and copying a shape. The scoring system was 0–2 points for each question, with a total score of 30 points. This interpretation was based on age and years of education. Higher scores indicate higher cognitive function. The participants were divided into normal cognitive function and suspected cognitive decline groups. The patients with MCI in this study were classified into the suspected cognitive decline group at the initial assessment. 

#### 2.3.2. Short Geriatric Depression Scale—Korean Version (SGDS-K) 

The SGDS-K is the Korean version of the Elderly Depression Scale [20] based on the Geriatric Depression Scale (GDS) [21]. It consists of 15 yes/no questions with a total score of 15 points. The SGDS-K assessed symptoms of depression over the past week. Examples of question are: “Are you generally satisfied with your current life?”, “Do you often feel that your life is boring?”, “Do you feel your memory is worse than other seniors your age?”. Higher scores indicated higher levels of depression. The high-risk depression group was represented by >10 points. The correlation of the SGDS-K was r = 0.95.

#### 2.3.3. Subjective Opinions 

Subjective opinions were collected using open-ended questions after the program. Questions were “How was the dementia prevention program?”, “What did you like in this program?”, “What was difficult?”, and “Is there anything you want from the program?”

### 2.4. Treatment

The online dementia prevention program was designed by two authors based on the APA Dementia Clinical Practice Guidelines. The guidelines include behavior, emotion, cognition, and stimulus-oriented approaches. The program consisted of 12 weekly sessions, including daily participation and occupational activities (Table 1). Participants performed daily tasks using a cognitive activity diary and a brain health playbook. Each participant performed the same set of weekly activities. The contents of the cognitive activity diary included writing down the date, weather, plan, medication of the day, and a coloring activity. Cognition training, such as memory activities, naming, and matching tasks were included in the brain health playbook. There were four sessions of stimulation-oriented approaches: making soap, planting, puzzles, and cooking activities.

### 2.5. Statistical Analysis

Descriptive statistical analyses (frequency, percentage, mean, and standard deviation) were conducted to obtain general information about the subjects. The effects of the program were evaluated using paired *t*-tests to compare the results before and after the program. Statistical significance was set at *p* < 0.05. Windows SPSS version 24.0 software was used for all analyses.

## 3. Results

### 3.1. Study Particpiation 

Table 2 presents the characteristics of the 101 participants. Females (83.2%), age in the 70s (57.4%), and 12–15 years of education (40.6%) were the most common. The average age of the 101 participants was 72.63 years (standard deviation = 4.95), range was 60 to 85 years old. The average number of years of education was 10.77 years (standard deviation = 3.65). Almost all of participants (98.0%) had more than 6–8 years of education. In Korea, 6–8 years of education means elementary to middle school. Sixteen participants (15.8%) had 16–20 years of education. This means a master’s degree or higher. Five participants (5.0%) were assessed for MCI by the CIST in the initial assessment.

### 3.2. Results of Cognitive Functions 

Based on the raw CIST score, orientation was maintained. Attention, visuospatial function, executive function, memory, and language function were improved (Table 3). Memory and total scores significantly improved after the program. Two of the five subjects with MCI based on the initial assessment returned to normal cognitive levels after treatment. In mean score, the change in total score was 0.88, change in memory was 0.48. There was no change in orientation.

### 3.3. Result of Depression 

The result indicated that symptoms of depression were significantly reduced from 3.35 ± 3.09 to 2.15 ± 3.06 (t = 4.49, *p* = 0.000) according to the SGDS-K after the dementia prevention program.

### 3.4. Subjective Opinions of Participants 

All participants answered open-ended questions based on their opinions of the program. As a result, subjective opinions were divided into four categories: (1) meaning in the activity itself, (2) cognitive stimulation, (3) reminiscence, and (4) challenges and achievements from the acquisition of new skills (Table 4). Some participants were very sad about the end of the program and wished to participate in the program again. Most of the participants (98.1%) would recommend this dementia prevention program to other community-dwelling older adults. 

## 4. Discussion

Continuous community participation and performance of meaningful daily tasks are important for maintaining and improving the cognitive function and mental health of community-dwelling older adults. During the COVID-19 pandemic, opportunities to participate in community activities and daily tasks have decreased significantly in South Korea. This study introduced a daily activity-based online dementia prevention program and investigated the program’s effectiveness on cognitive function and symptoms of depression of community-dwelling older adults. This program was designed based on the APA Dementia Clinical Practice Guidelines and consisted of daily tasks using a cognitive activity diary and a brain health playbook. The program gave community-dwelling older adults a chance to participate in daily tasks and to communicate with others. The program was effective in maintaining and improving their cognitive function and symptoms of depression. In particular, two out of five participants with MCI in the initial assessment improved to normal cognitive levels after the program. Therefore, a continuous dementia prevention program is essential for the community-dwelling older adults during the COVID-19 pandemic.

A comprehensive dementia prevention program is effective for older adults. The dementia prevention program in this study was designed comprehensively based on APA Dementia Clinical Practice Guidelines. In this study, we designed 12 online dementia prevention program sessions. The program consisted of four parts: cognitive, behavior, emotion, and stimulation approaches. Therefore, to design a dementia prevention program, clinicians should consider various areas to improve the cognitive function and emotional status of community-dwelling older adults. 

In the previous study, ten sessions of a non-face-to-face dementia prevention program were effective in twenty-six older adults (fifteen in the experimental group and eleven in the control group) in day and night care facilities [22]. Cognitive function and depression were assessed using the CIST and the SGDS-K scores as described in this study. Participants in the experimental group engaged with a non-face-to-face cognitive activity playbook for the older adults by watching videos. After the program, memory and language function items changed significantly. Moreover, the symptoms of depression improved. This is similar to the results of the present study where we found that memory and the symptoms of depression in community-dwelling older adults improved significantly. 

In another study, the effectiveness of a 14-week comprehensive cognitive training program in five older adults with mild dementia was investigated [14]. The comprehensive program in the previous study included memory training, motor movements, language activity, and daily life training. Especially, motor movement, verbal association, and categorization were included in the memory training program. In the activities of daily living (ADL) training, there were four occupational tasks including diary use. The use of a diary to write appointments in the previous study was similar to what was carried out in the present study. After treatment, the ADL and functional tasks improved significantly. Little improvement was observed in memory or psychiatric symptoms. This study supports the idea that even though participants were patients with dementia, daily cognitive activities and regular participation in meaningful activities such as ADL training are of great value in dementia treatment. In a previous study, diary use was a major activity in daily life training. Positive effects on cognition, depression, anxiety, and quality of life were also identified [14]. These results support the effectiveness of participation in daily tasks, such as diary use with cognitive training, in comprehensive dementia prevention programs. This is because daily tasks require a variety of cognitive factors, such as attention, orientation, sequencing, executive function, and memory. 

Participation in daily tasks effectively improves cognitive function and reduces depression. In the previous study, an 8-week daily task participation program improved daily life, cognitive ability, satisfaction, and orientation in forty-three subjects with early dementia [15]. Repetitive daily tasks using pencils and paper, calendars, personal memory notebooks, and mobile phones effectively improved orientation. In another study, twenty-four older adults participated in a diary-writing program three times per week for three months [16]. After participation in the program, cognitive and attention functions improved significantly. In previous studies, cognitive treatment and dementia prevention programs improved various aspects of cognition [14,15,16]. In this study, we observed significant effects in preventing memory decline and maintaining other cognitive functions, such as orientation, attention, visuospatial function, executive function, and language function, in healthy older adults without dementia. However, the results may vary as a result of the various subject groups, participants with diseases such as dementia, and different cognitive function assessment tools among the studies. Therefore, in-depth research is required to investigate the effects of dementia prevention programs on various aspects of cognitive function, ADL skills, and other mental health functions in different participant groups.

Social engagement and participation in daily tasks decreased symptoms of depression [11,14,22]. In previous studies, a dementia prevention and cognitive program decreased symptoms of depression [14,22]. In this study, daily task participation decreased the symptoms of depression in the healthy older adults. During the COVID-19 pandemic, a lack of social participation has been a major problem for community-dwelling older adults. Many community-dwelling older adults cannot participate in their usual community routines such as visiting senior community centers, sports teams, or public health centers. Based on the results of this study, daily task participation while keeping a diary or keeping in touch with occupational therapists could improve symptoms of depression in community-dwelling older adults. This result is similar to a previous finding that maintaining a diary reduced depressive symptoms in chronic rehabilitation patients and their caregivers [9]. Therefore, online programs are good substitutes. 

Many kinds of online rehabilitation and cognitive training programs have been introduced [7,23,24]. Online rehabilitation programs are effective for various functions such as cognition, motor function, ADL, and quality of life in older adults. In a previous study, language ability and episodic, verbal, and working memory significantly improved after three months of a remote cognitive rehabilitation program in twenty-seven individuals with mild memory impairment [23]. In another study, a comprehensive remote rehabilitation approach called ABILITY improved motor and cognitive skills in patients with early dementia [7]. Furthermore, in another previous study, thirty-one older adults with MCI and vascular cognitive impairment participated in a computer-based online rehabilitation program called GOAL Tele-R, which consisted of cognitive, physical, and social activities for 8 weeks [24]. They found that it was effective in preventing the deterioration of cognition, memory, and daily living functions. Therefore, the effectiveness of the comprehensive online dementia prevention program in this study is supported.

Online dementia prevention programs have several strengths over face-to-face programs. First, protection from infection is increased in online situations, which is important during the COVID-19 pandemic. Second, many people, especially those who suffer from locomotion difficulties, can participate in the same manner as in the videos at home. Third, participants can participate in the program anywhere and at any time, even if they live far away from a public health center. Finally, community-dwelling older adults can keep in touch with therapists weekly, so they can maintain their social relationships. In our opinion, these benefits have positive effects on cognition and depression among community-dwelling older adults.

However, difficulties in direct management and treatment, the need for trained helpers or facilitators, safety issues, and difficulties in checking treatment adherence are limitations of online dementia prevention programs. It is still necessary to visit in person when a detailed assessment or diagnosis is required. In this study, participants needed to travel twice for evaluation before and after treatment. Therefore, professionals should choose appropriate approaches depending on the settings or situations.

The strengths of this study were that it used a large sample size. Secondly, the effectiveness of an online dementia prevention program for normal community-dwelling older adults was investigated in community-based rehabilitation settings. Third, the program in this study was evidence-based, based on the APA Dementia Prevention Guidelines. Finally, the program was comprehensive, including cognition-, emotion-, behavior-, and stimulation-oriented factors. Therefore, we found great value in online dementia prevention programs for improving cognitive function and depression in community-dwelling older adults.

A limitation of this study is that generalization is difficult because the participants were recruited from one public health center. In addition, this study was not a randomized controlled study and there was no control group. Our program consisted of several programs. Therefore, determining which factors were the most influential is difficult. However, we found the comprehensive dementia prevention program used in our study to be effective for improving cognitive function and depression in community-dwelling older adults. Even though the authors checked the results of the participants weekly on the NAVER BAND apps, it was difficult to ascertain whether subjects watched the videos to the end.

In previous studies, effectiveness of the application of an online dementia prevention or cognitive treatment program for patients with MCI and dementia in various areas such as daily life, performance function, quality of life, and stress was identified. However, in this study, only cognitive function and symptoms of depression were assessed. Therefore, identifying the effects in various areas is necessary for future research. Furthermore, specialized online dementia prevention programs must be developed.

## 5. Conclusions

The online dementia prevention program was effective in improving cognitive function and symptoms of depression in healthy community-dwelling older adults and those older adults with MCI during the COVID-19 pandemic. Participation in meaningful daily occupational tasks, including keeping a diary and using a cognitive health playbook, is a useful method for improving cognitive function and decreasing symptoms of depression. Regular online communication with occupational therapists positively affects depression management. A dementia prevention program for community-dwelling older adults is important for community-based occupational therapy in Korea. Clinicians in community rehabilitation settings should be more interested in online dementia prevention programs and provide evidence-based programs.

## Figures and Tables

**Table 1 healthcare-11-01376-t001:** Contents of the Programs.

Approaches	Activities	Contents
▪ Cognition-oriented approaches Cognitive skill training ▪ Behavior-oriented approaches Scheduled habit training ▪ Emotion-oriented approaches Reminiscence activities	Structured cognitive activities diary	Write down the date, weather, plan, and medication Daily schedule arrangement Recalling morning cognitive activities Coloring activities
	Brain health playbook	Cognition training workbook Planning monthly and daily schedules Orientation, attention, memory, and executive function training
▪ Stimulation-oriented approaches Occupational activities Recreational activities Multisensory stimulation	Coloring book Making soap Planting Puzzle Cooking	Sensory: tactile, taste, smell Activities: fine motor, visuomotor, learning new skills Cognition: attention, memory, visuospatial, executive function, communication skills Psychosocial: psychological stability and sense of achievement

**Table 2 healthcare-11-01376-t002:** Status of Participants (N = 101).

	Variables	N	%
Gender	Male	17	16.8
	Female	84	83.2
Age	60–69	30	29.7
	70–79	58	57.4
	80–89	13	12.9
	Range	60–85	
	Average	72.63 ± 4.95	
Years of Education	0	1	1.0
	1–5	1	1.0
	6–8	22	21.8
	9–11	20	19.8
	12–15	41	40.6
	16–20	16	15.8
	Range	0–20	
	Average	10.77 ± 3.65	
Cognitive Function	Normal	96	95.0
	MCI	5	5.0

MCI, mild cognitive impairment.

**Table 3 healthcare-11-01376-t003:** Result of the Cognitive Impairment Screening Test (CIST).

Variables	Baseline	Post Treatment	Changes in Mean Score	t	*p*-Value
Orientation	4.89 ± 0.32	4.89 ± 0.47	0	0.00	1.000
Attention	2.59 ± 0.64	2.73 ± 0.68	0.14	1.82	0.070
Visuospatial function	1.90 ± 0.39	1.97 ± 0.30	0.07	1.54	0.127
Executive function	4.98 ± 0.97	5.03 ± 1.03	0.05	0.43	0.667
Memory	8.49 ± 1.84	8.97 ± 1.78	0.48	2.39	0.019 *
Language function	3.65 ± 0.52	3.73 ± 0.49	0.08	1.81	0.073
Total	26.45 ± 2.88	27.33 ± 2.76	0.88	3.55	0.001 *

** p* < 0.05.

**Table 4 healthcare-11-01376-t004:** Opinions of the Participants.

Categories	Individual Opinion
Meaning in the activity itself	During the COVID-19 pandemic, I could get away from the stuffy daily life and get more vitality. Raising and eating my own leek was a meaningful activity. It was fun to color without noticing the passage of time. It was nice to have something to do every day.
Cognitive stimulation	It was good for stimulating memory because I could check my activities while writing a diary. It was good to know my previous activities while looking at the diary. It was nice to be able to think about myself and routines.
Reminiscence	I felt like I was returning to my childhood by reminiscing. It was good for returning to the childhood of my children by doing the activities of coloring, diary writing, and keeping a brain health playbook.
Challenging achievement from acquisition of new skills	It was interesting to do new activities, such as soap making, puzzles, and finding texts. It was difficult to find the puzzle, but I enjoyed it after completing it. I was afraid to do something with the online program at first, but I got used to it and was able to do it well. It was difficult to use the app and upload photos, but it became easier as I did it. During the COVID-19, life was boring, but a lot of interesting things happened, so it was fun.

## Data Availability

Not applicable.

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
