# Peer review of "Effectiveness of an Online Dementia Prevention Program on Cognitive Function and Depression in Community-Dwelling Older Adults during the COVID-19 Pandemic in Korea"

_healthcare, 2023, doi:10.3390/healthcare11101376_

Round 1

Reviewer 1 Report

Introduction:

It might be helpful to consider replacing the phrase “non-face-to-face” to online or virtual mode. Personally, the use of phrase “non-face-to-face” is wordy and the main essence is that the authors are using internet to provide cognitive stimulation to participants remotely. The article examines the possible benefits of virtual or non-face-to-face dementia prevention programs for community-dwelling older adults. Some of the current sentences in the Introduction section are redundant. . The introduction section can be edited with inclusion of some prior literature and condensing of the current paragraphs into a single paragraph.

Methods:

·       Mild cognitive impairment is a different clinical diagnosis from dementia. It also needs to be separate from normal cognitive function. The current sentence on Page 2 Lines 80-81 makes it sound normal cognitive function includes MCI which is not accurate.

·       What was the duration of the program? How often were the weekly recordings made available to the participants?

·       When was data collected for this study? The entire world has gone through dramatic changes since the onset of COVID-19.  Clearly discussing the specific periods of data collection can also help explain the possible social isolation and challenges experienced by the participants and how this online cognitive stimulation program was a great change for these participants.

·       Page 3 Lines 98-99: Were participants with MCI considered to be in the normal cognitive function or suspected cognitive decline group?

·       How was the participation and hours of viewing by participants tracked or recorded?

·       Were all participants retired? Or were some of the participants working part-time or full-time?

·       Did all the participants complete the same set of activities and in the same exact sequence each week?

·       Were the participants tracked for any physical exercise or social stimulation they may have engaged in outside of these virtual cognitive stimulation programs? As the authors measured the possible changes in self-reported depression, it is possible that some of the other lifestyle choices and daily events may also have contributed to the improved mental health (as indicated by lower self-reported depression) at the end of the program.

Results:

·       Page 4 Lines 126-127: The current sentence is unclear. Maybe say 5 participants were assessed to have MCI based on CIST scores.

Discussion:

·       Page 6 Lines 193-194: What is meant by sub-programs? Did this include series of different activities each week?

·       What can the authors speculate about influence of other activities that may also have contributed to the improved self-reported depression status among the participants?

Please check the entire document for grammar and spelling errors.

Author Response

Your advice helped me complete the paper. We really appreciate it.

Reviewer 2 Report

English needs improvement, sometimes adverbs are used unnecessarily or wrongly. The line of thought is somewhat confusing, especially in the introduction and discussion. 

The article is about dementia prevention and meets the objectives of exposing the work done for that goal but the issue is focused on mobile health, it could be added some more information presenting some results or a comparison of satisfaction/adherence to treatment/ease of use between face to face and mobile treatments. With this kind of data, it will be possible to make a more complete and original discussion and a more appropriate conclusion to this specific issue.

I could not easily find the Naver band app, please add information about the app.

Author Response

(The authors gave the same response as above.)

Reviewer 3 Report

Kyoung-Chul Min etc. aimed to introduce evidence-based non-face-to-face dementia prevention programs in Korea and evaluated their effectiveness on cognitive function and depression in community-dwelling elderly. The study involved 101 community-dwelling elderly without dementia who participated in a 12-session non-face-to-face dementia prevention program designed by occupational therapists. Cognitive function and depression were assessed, and the results showed that non-face-to-face dementia prevention programs are an effective method to maintain and increase cognitive function and prevent depression in community-dwelling elderly during the COVID-19 pandemic. This paper provided sufficient background, methods are adequately described, and results are clearly presented.

Minor revision:

1.     In line 94, It said ‘It consists of 13 items including orientation, attention, visuospatial function, executive function, memory and language function.’ ‘13 items' is mentioned, however, it only includes 6 items. Please specify what are the 13 items. If you are only interested in the 6 items (or maybe it should be 6 domains), please rephrase to avoid the miss leading.

  1. From line 131 to line 132, it said ‘Two out of five subjects with MCI at the initial assessment returned to normal cognitive levels after treatment (Table 3).’ However, subjects with MCI can not be reflected in Table 3.  
  2. Table 3, suggests adding one column for the values of change from post treatment to baseline. Although the p-values from the t-test can provide significance, the values of change larger than 0 can more clearly show the improvement.  

Author Response

(The authors gave the same response as above.)
